# Colchicine Mutagenesis from Long-term Cultured Adventitious Roots Increases Biomass and Ginsenoside Production in Wild Ginseng (*Panax ginseng* Mayer)

**Kim-Cuong Le** [1,2]**, Thanh-Tam Ho** [3,4]**, Jong-Du Lee** [5]**, Kee-Yoeup Paek** [1,6] **and So-Young Park** [1,]*

[1] Department of Horticulture, Division of Animal, Horticultural and Food Sciences, Chungbuk National University, Cheongju 28644, Korea; lekimcuong88@gmail.com (K.-C.L.); paekky@chungbuk.ac.kr (K.-Y.P.)

[2] Department of Forest Genetics and Plant Physiology, Swedish University of Agricultural Sciences, 901 83 Umea, Sweden

[3] Institute for Global Health Innovations, Duy Tan University, Da Nang 550000, Vietnam; hothanhtam2@duytan.edu.vn

[4] Faculty of Pharmacy, Duy Tan University, Da Nang 550000, Vietnam

[5] Biodiversity Research Institute, Shinyedong-ro 338, Seogwipo-si, Jeju 63608, Korea; zpzp011@nate.com

[6] WellGreen Co., Chungdae-ro 1, Cheongju 28644, Korea

\* Correspondence: soypark7@cbnu.ac.kr; Tel.: +82-43-261-2531

**Abstract:** *Panax ginseng* Mayer is a perennial herb that has been used as a medicinal plant in Eastern Asia for thousands of years. The aim of this study was to enhance root biomass and ginsenoside content in cultured adventitious roots by colchicine mutagenesis. Adventitious *P. ginseng* roots were treated with colchicine at different concentrations (100, 200, and 300 mg·L$^{-1}$) and for different durations (1, 2, and 3 days). Genetic variability of mutant lines was assessed using random amplification of polymorphic DNA (RAPD) analysis. Ginsenoside biosynthesis gene expression, ginsenoside content, enzyme activities, and performance in bioreactor culture were assessed in four mutant lines (100–1-2, 100–1-18, 300–1-16, and 300–2-8). The results showed that ginsenoside productivity was enhanced in all mutant lines, with mutant 100–1-18 exhibiting the most pronounced increase (4.8-fold higher than the control). Expression of some ginsenoside biosynthetic enzymes was elevated in mutant lines. Enzyme activities varied among lines, and lipid peroxidation activity correlated with root biomass. All four lines were suitable for bioreactor cultivation, with mutant 100–1-18 exhibiting the highest biomass after culture scale-up. The results indicated that colchicine mutagenesis of *P. ginseng* roots increased biomass and ginsenosides production. This technique, and the root lines produced in this study, may be used to increase industrial yields of *P. ginseng* biomass and ginsenosides.

**Keywords:** adventitious roots; biomass; colchicine; ginsenosides; mutagenesis

## 1. Introduction

Cultivation of plant cell tissues and organs is one of the most powerful tools available for research into plant propagation, crop development, production of valuable phytocompounds, and preservation of endangered plant species. However, in vitro plant materials accumulate genetic and epigenetic changes at high frequencies during long-term culture. Long periods of continuous subculture also enhance somaclonal variation [1], which leads to changes in phenotypes such as plant size, yield,

and disease and insect tolerance in ornamentals [2,3], and changes to secondary metabolite production in medicinal plants [2,4–7]. Somaclonal variation can lead to critical economic losses when plant cell and organ culture is used for biomass propagation and uniform plant germination [1].

*Panax ginseng* Mayer (Araliaceae), known as ginseng, is one of the most important oriental herbs worldwide. Cell suspension, adventitious root, and hairy root culture systems were established for ginseng at Chungbuk National University around 2000 [8,9]. However, production of the primary bioactive compounds, ginsenosides (triterpenoid saponins), decreased after long-term continuous subculture of *P. ginseng*. Kiselev et al. [7] showed that ginsenosides comprised only approximately 0.024% dry weight of a 20 years cell subculture. Similarly, Li et al. [6] reported that paclitaxel was undetectable after five years of *Taxus chinesesis* cell subculture, and anthocyanin and alkaloid contents in *Vitis vinifera* and *Catharanthus roseus*, respectively, were gradually lost during long-term culture [4,5].

Several biotechnological technologies, such as primary culture, induction of polyploidy, and mutagenesis, can be used to overcome the difficulties encountered during long-term subculture. However, ginseng plants collected from their natural environment can be expensive, depending on their age, and ginseng, particularly wild ginseng, is rare in nature. Primary culture methods are thus inappropriate for such endangered or perennial plant materials. Colchicine is an alkaloid derivative from amino acids phenylalanine and tyrosine of *Colchicum autumnale*. The DNA breakage and clastogenicity of alkylating agents in plants has been studied for over 40 years [10]. Colchicine, which has been used for a long time to induce polyploidy in higher plants by inhibition of spindle fiber formation [11–13], is also an effective mutagen in many plant species [14–20]. Mutagenesis induction techniques combined with in vitro culture and molecular marker systems have facilitated the development of genetic variability platforms to produce enhanced reproductive and vegetative organs [17,21,22]. Significant mutagenic effects of colchicine result in the physiological disturbances and chromosome aberrations by alterations of the signaling pathway [17,20,23]. In medicinal plants, the selection of an efficient mutagenic agent is critical for generation of profitable and desired mutations at high frequency. Mutant induction using colchicine was previously used successfully in a range of plants, such as *Hordeum vulgare*, *Helianthus annuus*, *Sesame indicum*, *Calendula officinalis*, *Peniesetum purpureum*, *Glycine max* and *Bacopa monnieri* [15–20,24]. However, very few studies report the enhancement of secondary metabolite production in *P. ginseng* after mutagenesis with colchicine.

In the present study, colchicine mutagenesis of wild ginseng produced mutant adventitious roots with enhanced root biomass and ginsenoside content. DNA polymorphisms in the mutant lines were analyzed using random amplification of polymorphic DNA (RAPD) markers, and expression of genes critical for ginsenoside biosynthesis (*PgSS*, *PgSE$_2$*, *PPDS*, and *PPTS*) was assessed. Mutant lines were also assessed for their culture characteristics in flask and bioreactor cultivation. Finally, for the first time a correlation between Malondialdehyde (MDA) activity/biomass and adaptation capability in mutant roots was established.

## 2. Materials and Methods

### 2.1. Plant Materials

Adventitious roots derived from calluses of 100-year-old wild ginseng (*P. ginseng* Mayer) roots were cultured in MS medium [25] supplemented with indole-3-butyric acid (IBA) and 3% (*w/v*) sucrose. The cultures were maintained at 24 ± 1 °C in the dark.

### 2.2. Mutant Root Induction with Colchicine

MS liquid medium supplemented with different concentrations (0, 100, 200, and 300 mg·L$^{-1}$) of filtered-sterilized colchicine (Sigma-Aldrich, St. Louis, MO, USA) was utilized for mutant root induction. Ten adventitious roots (1.5 cm length) were utilized per replicate, with four replicates per treatment. Adventitious roots were placed in 100 mL flasks containing 20 mL of MS liquid medium and colchicine. The flasks were incubated on a rotary shaker (100 rpm) for 1, 2, and 3 days in darkness.

Root tips, 1.5 cm long, were washed three times with MS medium and then transferred to fresh solid medium supplement with 5.0 mg·L$^{-1}$ IBA, 3.0% sucrose and cultured in the dark at 24 ± 1 °C. Lateral root induction, the number of lateral roots per explant and lateral root length were recorded after eight weeks. The lateral roots of treated roots from each treatment were detached and cultured on the same medium describe above. The putative mutant roots were firstly screened for morphology and biomass after six weeks of culture. The four putative lines with the highest biomass were selected and screened again for ginsenoside content.

Mutant roots were transferred to a flask containing 100 mL of MS liquid medium containing IBA and 3% sucrose and assessed in a scaled-up 3 L bioreactor containing 2 L of MS liquid medium. The inoculation density of root in flask and bioreactor culture was 5.0 mg·L$^{-1}$. Lateral root number, root length, and biomass (fresh and dry weight) were measured in the flask and bioreactor cultures. Mutant root morphology was also examined.

### 2.3. DNA Content Analysis by Flow Cytometry

DNA content of colchicine-treated adventitious roots was analyzed by flow cytometry (Partec PA, Münster, Germany) after six weeks of culture. Three to four lateral roots were chopped using a razor blade in 0.4 mL nuclei extraction buffer (CyStain UV Precise P, Partec, Münster, Germany) for 30 s. The sample was incubated for 5 min and filtered through a Partec 50 µm celltrics®disposable filter. Staining buffer (1.6 mL) was added to the sample tube, incubated for 30 s and analyzed in a flow cytometer. At least 3000–6000 nuclei were measured per sample and processed using FloMax software (Partec, Münster, Germany). The peak of nuclei insulated from a tetraploid (control) lateral root of *P. ginseng* was adjusted at channel 100.

### 2.4. RAPD Analysis

#### 2.4.1. DNA Isolation

Fresh 6-week-old adventitious roots (0.1 g) from selected putative mutant lines and from three tetraploid control lines were homogenized using a Tissuelyser II (Qiagen, Hilden Germany). Genomic DNA was isolated following the cetyl trimethyl ammonium bromide (CTAB) extraction method [26]. CTAB buffer solution and 0.2% β-mercaptoethanol were added to the homogenized adventitious roots samples. Samples were incubated at 65 °C for 30 min, chloroform:isoamyl alcohol extraction solution (24:1) was added, and DNA was precipitated with isopropanol. Pellets were washed with 70% ethanol and resuspended in 50 µL of sterilized water. DNA concentration was measured using a DS-11 + spectrophotometer (Denovix, Inc., Wilmington, DE, USA).

#### 2.4.2. PCR Amplification

Amplification was performed using a C1000TM Thermal Cycler (CFX96 Touch™ Real-time PCR Detection System, Bio-Rad Laboratories, Inc., Singapore) as described by Williams et al. (1990). Twenty oligonucleotides from Operon Inc (USA) were used for initial PCR amplifications, and six primers (A-05, A-08, A-09, A-10, A-13, and A-15) were subsequently selected for further amplifications (Table 1). PCR was conducted in a total volume of 20 µL containing DNA template (10 ng·µL$^{-1}$), Primer Tag Premix (2×) (Genetbio Inc., Daejeon, Korea), primers, and sterilized water. RAPD amplification was performed over 35 cycles as follows: 94 °C for 1 min, 37 °C for 30 s, 72 °C for 1 min, and 72 °C for 5 min, after which the reactions were held at 4 °C. PCR products were analyzed using 2% agarose gel electrophoresis in 1× TEB buffer. Electrophoresed gels were examined with UV transillumination.

**Table 1.** Nucleotide sequences of RAPD primers that detected polymorphism. RAPD: random amplification of polymorphic DNA.

| Primers | Sequence |
| --- | --- |
| A-05 | 5′-AGGGGTCTTG-3′ |
| A-08 | 5′-GTGACGTAGG-3′ |
| A-09 | 5′-GGGTAACGCC-3′ |
| A-10 | 5′-GTGATCGCAG-3′ |
| A-13 | 5′-CAGCACCCAC-3′ |
| A-15 | 5′-TTCCGAACCC-3′ |

*2.5. RNA Isolation and Quantitative Real-Time PCR Analysis of Gene Expression*

Total RNA (100 mg) was isolated from mutant roots after 6 weeks of culture using NucleoZol reagent (MACHEREY-NAGEL GmbH& Co. KG, Düren, Germany). Total RNA was incubated at 65 °C for 5 min and reverse transcribed using ReverTra Ace®qPCR RT Master (TOYOBO CO, LTD, Osaka, Japan) in 10 μL reaction volumes containing 1 μg of total RNA and 5× RT Master (ReverTra Ace®, RNase inhibitor, oligo dT primer, random primer, MgCl2 and dNTPs). Reverse transcription reactions were performed at 37 °C for 15 min, and then 50 °C for 5 min. The mixture was then heated at 98 °C for 5 min followed by cooling at 4 °C to terminate the reaction. For quantitative real-time PCR (qPCR), specific primers were designed using the Primer3Plus program (http://carbon.bioneer.co.kr/primer3plus/). The primers used were *P. ginseng* ACT1 (actin, GenBank accession no. KF699319), *P. ginseng* SS2 (squalene synthase, GenBank accession no. GQ468527), *P. ginseng* SE2 (squalene epoxidase, GenBank accession no. FJ393274), *P. ginseng PPDS* (cytochrome P450 CYP716A47, GenBank accession no. JN604537), and *P. ginseng PPTS* (cytochrome P450 CYP716A53v2, GenBank accession no. JX036031) (Table S1). Gene expression levels of *PgSS*, $PgSE_2$, *PPDS*, and *PPTS* in *P. ginseng* mutant roots were qPCR quantified using a C1000TM Thermal Cycler (CFX96 Touch™ Real-time PCR Detection System, Bio-Rad Laboratories, Inc., Singapore, Singapore) with the following cycle: 95 °C for 30 s; then 40 cycles of 95 °C for 5 s, 55 °C for 10 s, and 72 °C for 15 s; and a final 10 min extension at 72 °C.

*2.6. Preparation and Extraction of Mutant Lines*

The 6-week-old root samples (0.5 g dry weight) of control and selected mutant roots were placed in 100 mL flasks with 50 mL of 80% (v/v) ethanol. Flasks were sonicated in an ultra-sonication bath (SD-D250H by Mujigae Co., Seoul, Korea) for 1 h at room temperature. Thereafter, extracts were filtered through Whatman filter paper (No. 1002 110) and collected in a round-bottom flask. The solvent was evaporated to dryness using a rotary evaporator (N-1000, Eyela, Tokyo, Japan) at 40 °C, and the residue was dissolved in 50 mL of distilled water. The aqueous solution was washed twice with 50 mL of ethyl ether, and the aqueous layer was then extracted twice with 50 mL of water-saturated n-butanol. The n-butanol fraction was evaporated using a rotary evaporator at 50 °C. The sample solution was dissolved in 2 mL of methanol and filtered through a 0.2 μm Whatman syringe filter before analysis by High Performance Liquid Chromatography (HPLC).

*2.7. HPLC Analysis of Ginsenosides*

Extracted samples were analyzed by HPLC using a Waters 2695 separation module with a 2996 photodiode array detector on a Fortis C18 column (φ 5 μm, 150 × 4.6 mm). The mobile phase comprised acetonitrile (solvent A) and water (solvent B). The following gradient elution formula was used: 0 min, 18% A, 82% B; 0–42 min, 24% A, 76% B; 42–46 min, 29% A, 71% B; 46–75 min, 40% A, 60% B; 75–100 min, 65% A, 35% B; 100–135 min, 85% A, 15% B; and 135–150 min, 85% A, 15% B. A flow rate of 0.6 mL·min-1 was maintained throughout. Ginsenosides were detected at 203 nm. The sample injection volume was 20 μL, and the temperature of the column was controlled at 35 °C. Ginsenoside standards were purchased from ChromaDex (USA). Protopanaxadiol-type (Rb1, Rb2, Rb3, Rc, Rd, Rg3, Rh2) and protopanaxatriol-type (Re, Rf, Rg1, Rh1) standard solutions were used.

Ginsenoside content was calculated as follows: Ginsenoside content (mg·g$^{-1}$ DW) = ginsenoside concentration in sample (mg·L$^{-1}$) × sample volume (L)/root dry weight (g). Total ginsenosides content was calculated as the sum of the ginsenoside fractions. Ginsenoside productivity was calculated as follows: Ginsenoside productivity (mg·L$^{-1}$) = Total ginsenoside content (mg·g$^{-1}$ DW) × harvested root dry weight (g)/volume of culture medium (L).

*2.8. Lipid Peroxidation*

Lipid peroxidation contents were measured using a modified malondialdehyde (MDA) method [27]. The 6-week-old root tissue (0.4 g) of control and selected mutant roots were disrupted in liquid nitrogen with a mortar and pestle and homogenized in 0.1% trichloroacetic acid (TCA). The homogenate was centrifuged at 5000 rpm for 5 min. The supernatant was then blended with 0.6% TBA and placed in a water bath at 90 °C for 30 min. Absorbance was measured at 532 and 600 nm using a spectrophotometer (Optizen POP, Mecasys Co., Ltd, Daejeon, Korea).

*2.9. Catalase and Peroxidase Activity*

Fresh 6-week-old adventitious root tissue (1.0 g) of control and mutant root lines (100–1-2, 100–1-18, 300–1-16, and 300–2-8) were disrupted in liquid nitrogen and homogenized in 4 mL of extraction buffer (50 mM potassium phosphate buffer, 2.0% polyvinylpolypyrrolidone, and 1.0 mM phenylmethylsulfonylfluoride). The extracted sample was centrifuged at 13,000 rpm for 10 min at 4 °C (Smart R17, Hanil science CO., LTD, Gimpo, Korea). The supernatant was stored at 2 °C and used for enzyme assays within 4 h.

*Catalase*, CAT (EC 1.11.1.6) activity was estimated in a reaction mixture containing 500 µmol H$_2$O$_2$ in 10 mL of 100 mM phosphate buffer (pH 7.0). CAT activity was determined by monitoring H$_2$O$_2$ consumption at 240 nm for 3 min. The results were expressed as CAT unit mg$^{-1}$ protein (1 mM of H$_2$O$_2$ reduction min$^{-1}$·mg$^{-1}$ protein) [28].

*Peroxidase*, POD (EC 1.11.1.7) activity was measured using a modified method described by Bisht et al. [28]. The reaction mixture contained 1.5 mL of phosphate buffer (pH 7.0), 1% guaiacol, and 1% H$_2$O$_2$. POD activity was assessed as the change in absorption at 470 nm as a result of guaiacol oxidation. The results were expressed as unit min$^{-1}$·mg$^{-1}$ protein.

*2.10. Statistics*

RAPD data were scored for the absence "0" or presence "1" of electrophoresis band. Scoring data were entered into a binary matrix and analyzed using NTSYSpc 2.1. Similarities were calculated using Jaccard's coefficient, and a dendrogram was established using Unweighted Pair Group Method with Arithmetic Mean (UPGMA). Data were analyzed using SPSS version 16.0 (SPSS Inc., Chicago, IL, USA). Where a significant difference ($p < 0.05$) was observed for a measured parameter, means were separated using Duncan's multiple range test at the 5% level.

## 3. Results and Discussion

*3.1. Effect of Colchicine Treatment on Lateral Root Induction and Growth*

The induction of secondary roots after exposure to different concentrations of colchicine (100, 200, and 300 mg·L$^{-1}$) for different durations (1, 2, and 3 days) was estimated eight weeks after treatment (Table 2; Figure 1). All the colchicine-treated adventitious root explants survived until eight weeks after treatment (data not shown). Colchicine treatment appreciably affected the number of explants with induced lateral roots (Table 2). Treatment slowed the induction and branching of lateral roots during the initial 2–4 weeks of culture (data not shown). However, there was no significant difference in lateral root formation between control and treated roots after eight weeks of cultivation (Table 2). It was possible that the colchicine-treated explants exhibited slowed root induction as a result of a physiological disturbance that reduced the cell division rate and caused initial growth

retardation [29]. Colchicine-treated adventitious roots produced fewer, shorter lateral roots than untreated roots (Table 2). This may have been due to the loss of microtubules and the occurrence of sticky supercoiled chromosomes and c-mitoses. Loss of cortical microtubules likely resulted in cell expansion rather than cell elongation, producing shorter and thicker lateral roots after colchicine treatment (Figure 1C,D,G,H) [29,30]. Obute et al. [23], and Hewawasam et al. [31] observed similar suppression of growth in *Vigna unguiculata*, *Cucumeropsis mannii*, and *Crossandra infundibuiformis*. The presence of IBA in culture medium might have counteracted the initial harmful effects of colchicine [29]. New adventitious roots were detached, cultured, and screened by biomass. Four biomass-enhanced root lines were selected, and the histogram of nuclei (Figure. S1) and DNA index (Table 3) were assessed using flow cytometry, which showed that all four lines and controls were tetraploid. The capacity of colchicine to infiltrate inside cell of living organisms to interact with the DNA produces the typical toxic effects related to colchicine properties. Thereby, the mutagenic effects are principally caused by the direct interaction between the mutagen and DNA molecules [10,17].

**Table 2.** Effect of colchicine treatments on DNA content and growth in adventitious root of *Panax ginseng*.

| Colchicine Concentration (mg·L$^{-1}$) | Exposure Time (day) | Lateral Root Formation (%) | No. of Lateral Roots/Explant | Lateral Root Length (cm) |
|---|---|---|---|---|
| 0 | 0 | 100.0a $^z$ | 27.75a | 2.51a |
| 100 | 1 | 100.0a | 25.75b | 2.24c |
|  | 2 | 100.0a | 20.13c | 1.79e |
|  | 3 | 100.0a | 12.25g | 1.50g |
| 200 | 1 | 100.0a | 18.25d | 2.31b |
|  | 2 | 100.0a | 15.00e | 2.01d |
|  | 3 | 97.5ab | 11.13h | 1.60f |
| 300 | 1 | 100.0a | 17.50d | 2.34b |
|  | 2 | 97.5ab | 13.38f | 2.05d |
|  | 3 | 95.0b | 10.75h | 2.01d |

$^z$ Different letters within a column indicate significant difference at $p < 0.05$ according to Duncan's multiple range test ($n = 4$).

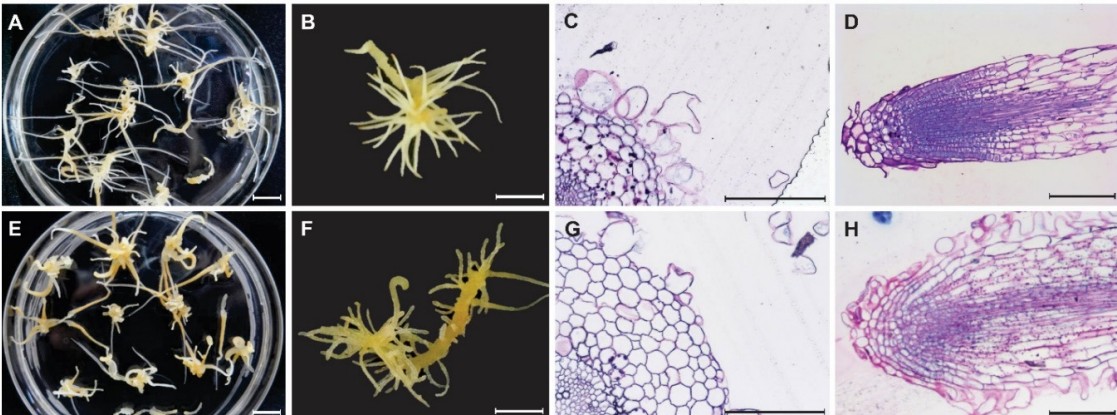

**Figure 1.** Morphology (**A**,**B**,**E**,**F**) and histology (**C**,**D**,**G**,**H**) of control and mutant root (100–1-18) in *Panax ginseng* after six weeks of culture. Adventitious root of control (**A**) and mutant 100–1-18 (**E**). Close-up adventitious root of control (**B**) and mutant 100–1-18 (**F**). Cross section of root (**C**,**G**), and longitudinal section of primordia (**D**,**H**) of control (**C**,**D**) and mutant root (100–1-18) (**G**,**H**) in *Panax ginseng*.

The four putative lines with the highest biomass among the total of 254 lines after screened by morphology and biomass (data not showed) were selected and screened again for ginsenoside content. Putative mutant roots were propagated in 250 mL flasks containing 100 mL of MS medium supplemented with 5 mg·L$^{-1}$ IBA and 3% sucrose, in the dark, on a rotary shaker (100 rpm) (Table 3).

The putative mutant roots were initially screened for high root biomass after six weeks of culture. Further experiments using DNA marker techniques were used to elucidate the genetic backgrounds of the four selected lines.

Four mutant root lines (100–1-2, 100–1-18, 300–1-16, and 300–2-8) exhibited higher growth ratios and more rapid growth than the control. Lateral root number, root length, and fresh and dry biomass were also recorded in the four lines after six weeks of culture (Table 3). The mutant roots (Figure 1E,F,G,H) were thicker and formed more abundant and longer lateral roots than the control roots (Figure 1A,B,C,D). The biomass obtained after six weeks of culture was approximately 1.5-fold higher for mutant roots than for control roots (Table 3).

**Table 3.** Characteristics of mutant roots of *Panax ginseng* after six weeks of flask and bioreactor culture.

| Culture Scale | Lines | DNA Index $^z$ | No. of Lateral Roots/Explant | Fresh Mass $(g \cdot L^{-1})$ | Dry Mass $(g \cdot L^{-1})$ | Growth Ratio $^y$ |
|---|---|---|---|---|---|---|
| Flask (250 mL) | Control | 1.00a | 24.00c $^x$ | 70.08b | 4.94b | 6.01b |
| | 100–1-2 | 1.08a | 32.00bc | 111.32a | 8.69a | 10.14a |
| | 100–1-18 | 0.99a | 38.25ab | 116.27a | 9.95a | 10.63a |
| | 300–1-16 | 1.00a | 28.50bc | 118.60a | 8.14ab | 10.86a |
| | 300–2-8 | 0.96a | 48.00a | 110.71a | 8.45a | 10.07a |
| Bioreactor (3 L) | Control | 1.00a | 25.83c | 73.36d | 5.39c | 13.91c |
| | 100–1-2 | 1.08a | 39.83b | 84.31c | 6.25bc | 16.10b |
| | 100–1-18 | 0.99a | 50.50a | 94.87b | 7.19ab | 17.98a |
| | 300–1-16 | 1.00a | 39.50b | 93.28b | 7.60ab | 17.66a |
| | 300–2-8 | 0.96a | 38.50b | 99.84a | 6.87ab | 18.98a |

$^z$ DNA index = (DNA value of mutant line $G_0/G_1$)/(DNA value of control $G_0/G_1$). $^y$ Growth ratio = [Harvested dry weight (g) − Inoculated dry weight (g)]/Inoculated dry weight (g). $^x$ Different letters within a column indicate significant difference at $p < 0.05$ according to Duncan's multiple range test ($n = 4$).

## 3.2. RAPD Analysis of Mutant Roots

RAPD analysis was used to assess genetic differences between the control and four putative mutant lines because it is a rapid detect method that does not require prior information regarding the nucleotide sequence, is inexpensive, and is suitable for detecting DNA alternations after exposure with mutagenic agents [32,33]. Six of eighteen RAPD primers (Table 1) produced different band patterns between control and mutant lines (data not shown). The principal differences seen in the RAPD profiles were the presence or absence of varying bands. Amplification data from the differential primers were scored and used to assess genetic distance (Figure 2A). Genetic distances between the control and the four mutant lines were 0.46–1.0, as assessed using Jaccard's coefficient matrix (Figure 2B). UPGMA analysis of RAPD data assigned the control to one cluster and the four mutant lines to a separate cluster. Of the four mutant lines, 100–1-18 differed substantially from the other three lines. Although RAPD analysis is most commonly used for phytogenetic, taxonomic, and genetic mapping studies, RAPD can also be used for detection of DNA damage and mutation [34]. In the present study, RAPD markers were effective for analysis of *P. ginseng* mutant adventitious root lines.

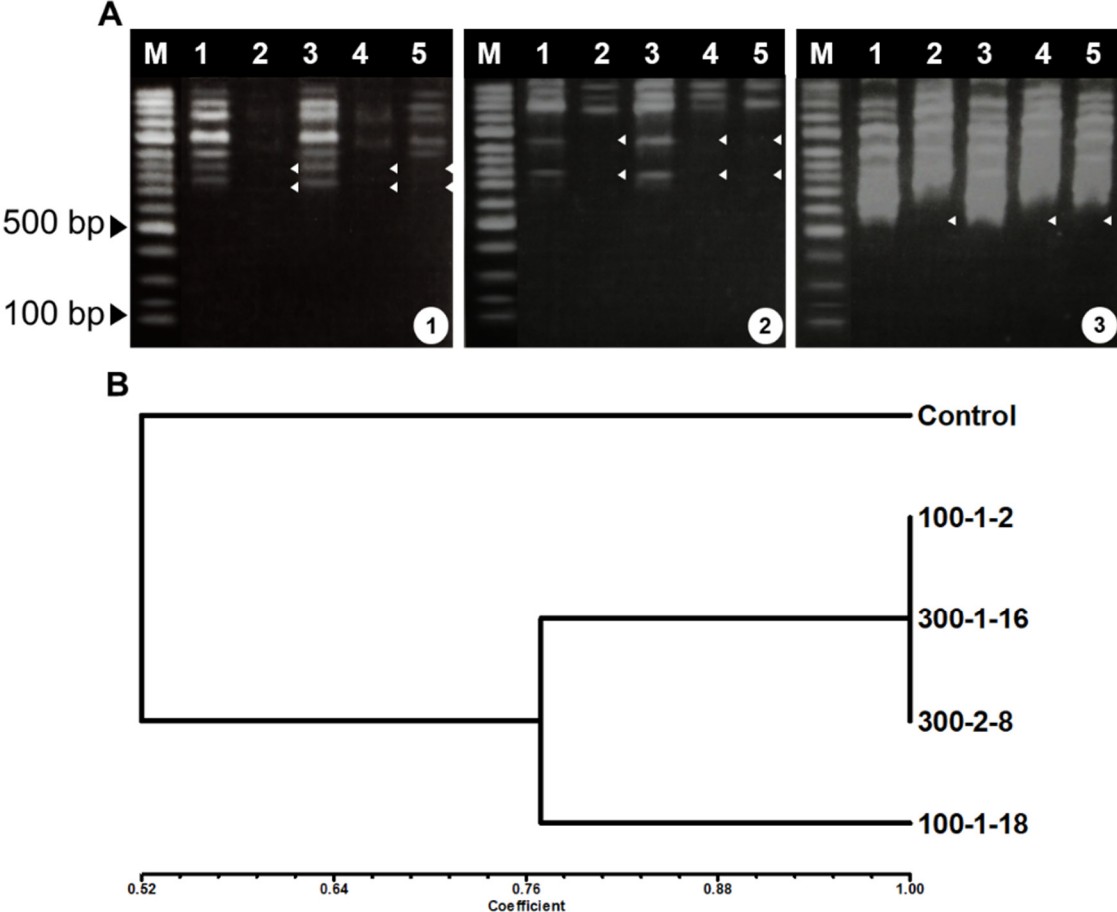

**Figure 2.** RAPD amplification (**A**) and similarity coefficient dendrogram (**B**) of mutant adventitious roots of *Panax ginseng*. Primers $A_1$: A-05, $A_2$:A10, and $A_3$:A-15),. M, marker; lane 1, control; lane 2, 100–1-2; lane 3, 100–1-18; lane 4, 300–1-16; and lane 5, 300–2-8.

*3.3. Expression Analysis of Ginsenoside Biosynthetic Genes*

The expression levels of four important genes involved in ginsenoside synthesis (*PgSS*, *PgSE₂*, *PPDS*, and *PPTS*) were analyzed to better understand the molecular characteristics of the four mutant lines (Figure 3). Triterpene ginsenosides are principally biosynthesized through the mevalonic acid pathway in the cytoplasm and the methylerythriol phosphate pathway in the chloroplast [35,36]. Squalene synthase (*PgSS*) catalyzes the biosynthesis of triterpenes [37], and squalene epoxidase (*PgSE₂*) is involved in the production of 2,3-oxidosqualene [38]. Cytochrome P450 enzymes are involved in a further stage of ginsenoside synthesis. *PPDS* (CYP716A47) catalyzes the formation of protopanaxadiol (PPD) from drammarenediol-II, and PPTS (CYP716A53v2) catalyzes the formation of protopanaxatriol (PPT) from PPD. Differences in *PgSE₂*, *PPDS*, and *PPTS* expression were observed among mutant and control lines. *PgSS* transcription was elevated in the four mutants compared with the control. *PgSS*, *PgSE₂*, *PPDS*, and *PPTS* were highly expressed in mutant 100–1-18 (Figure 3). The *PgSE₂* gene was also highly expressed in mutant 300–2-8, whereas the *PgSE₂*, *PPDS*, and *PPTS* genes were minimally expressed in mutants 100–1-2 and 300–1-16 (Figure 3). The results clearly demonstrated that ginsenoside biosynthesis genes were expressed at higher levels in the mutant 100–1-18 than in the other mutant lines.

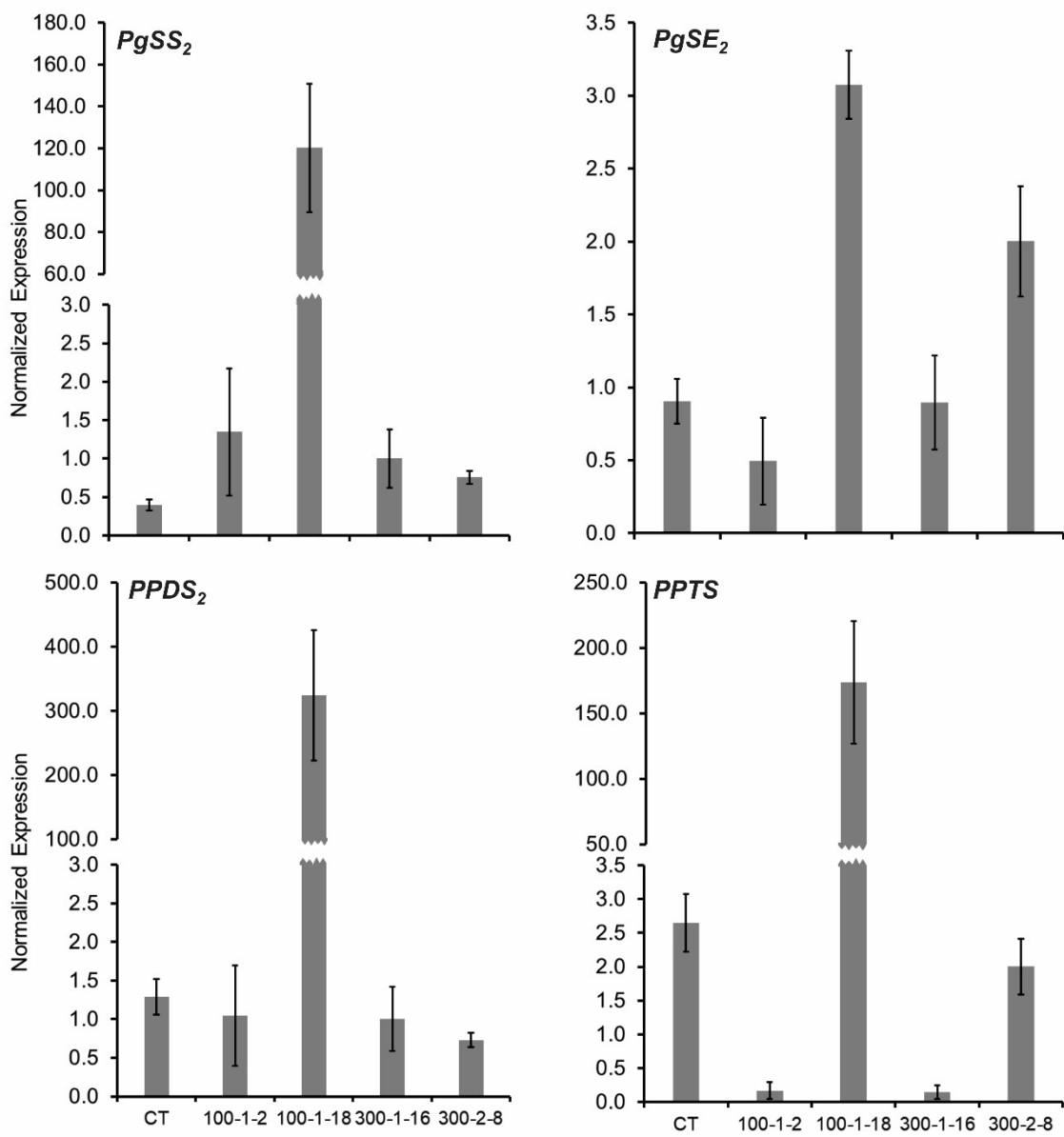

**Figure 3.** Gene expression levels of four mutant *Panax ginseng* root lines. *SS*, squalene synthase; *SE*, squalene epoxidase; *PPDS*, cytochrome P450 CYP716A47; and *PPTS*, cytochrome P450 CYP716A53v2.

*3.4. Ginsenoside Content Analysis by HPLC*

Ginsenoside content was determined in the four mutant lines after six weeks of culture. The HPLC chromatogram with the retention time values of the determined ginsenosides and their standards are shown at Table S2 and Figure S2. The total contents of 11 ginsenosides increased in all mutant lines compared with the control (Table 4). Mutant 100–1-18 produced the highest amounts of total PPD and PPT ginsenosides (2.5-, 2.12-, and 2.68-fold higher than the control, respectively) (Table 4). In the PPD group, the Rb1, Rb2, and Rc contents of 100–1-18 were 3.7-, 4.1-, and 2.3-fold higher than the control, respectively (Table 4). In the PPT group, the Re, Rf, Rg1, and Rg2 contents of 100–1-18 were 2.87-, 4.7-, 1.7-, and 3.2-fold higher than the control, respectively (Table 4). The PPD/PPT ratio differed among mutant and control roots: the 100–1-2 roots exhibited the highest PPD/PPT ratio, and the 300–2-8 roots had the lowest ratio. Overall, mutant 100–1-18 had the highest ginsenoside productivity (186.6 mg·L$^{-1}$), which was 4.85-fold higher than that of the control. Ginsenoside contents differed among the mutant and control lines as a result of disparate expression of ginsenoside synthesis genes. The high accumulation of ginsenosides in mutant 100–1-18 corresponded with elevated expression

of *PgSE₂*, *PPDS*, and *PPTS* (Figure 3; Table 4). The increased accumulation of ginsenoside content in mutant lines indicated that colchicine had a pronounced effect on ginsenoside synthesis. Mutants with elevated biomass and increased production of effective compounds are particularly valuable for medicinal plants [20,39]. Kharde et al. [20] reported that bacoside production increased 4-fold in colchicine-treated in vitro *Brahmi* plants. In this study, mutant roots exhibited higher ginsenoside contents than control roots, demonstrating that this phytochemical characteristic was strongly affected by mutagenesis (Table 4). Mutant roots exhibited thicker and longer growth than control roots, which may have contributed to the elevated ginsenoside levels. Further studies are needed to fully understand the molecular mechanisms governing high ginsenoside content and productivity in *P. ginseng*. The ginsenoside content of mutant line 100–1-18 produced the highest PPD (Rb1, Rb2, Rg3, and Rh2) and PPT (Re, Rg1, and Rg2) levels. The 100–1-18 line is therefore of possible commercial pharmacological value.

**Table 4.** Ginsenoside profiling of mutant *Panax ginseng* roots by HPLC analysis.

| Line | PPD (mg·g⁻¹ DW) | | | | | | | PPT (mg·g⁻¹ DW) | | | | PPD (mg·g⁻¹ DW) | PPT (mg·g⁻¹ DW) | PPD/PPT | Total Ginsenoside (mg·g⁻¹ DW) [z] | Ginsenoside Productivity (mg·L⁻¹) |
|---|---|---|---|---|---|---|---|---|---|---|---|---|---|---|---|---|
| | $Rb_1$ | $Rb_2$ | $Rb_3$ | Rc | Rd | $Rg_3$ | $Rh_2$ | Re | Rf | $Rg_1$ | $Rg_2$ | | | | | |
| Control | 1.08d [y] | 0.29c | 1.23a | 0.39b | 0.24a | 0.06a | 0.52a | 1.64c | 0.07b | 0.62b | 0.22b | 3.71b | 2.54c | 1.46b | 6.25b | 38.45c |
| 100–1-2 | 2.98b | 0.51b | 0.15b | 0.59b | 0.46a | 0.06ab | 0.53a | 1.13c | 0.09b | 0.53b | 0.18b | 5.08b | 1.93c | 2.63a | 7.01b | 77.21b |
| 100–1-18 | 4.01a | 1.19a | 0.28b | 0.88a | 0.50a | 0.05a | 0.48a | 4.72a | 0.33a | 1.06a | 0.71a | 7.85a | 6.81a | 1.15c | 14.66a | 186.60a |
| 300–1-16 | 2.18bc | 0.52b | 0.14b | 0.35b | 0.30a | 0.05ab | 0.44a | 1.60c | 0.05b | 0.64b | 0.25b | 4.22b | 2.54c | 1.66b | 6.76b | 72.60b |
| 300–2-8 | 1.74cd | 0.45bc | 0.10b | 0.29b | 0.31a | 0.10b | 0.44a | 2.29b | 0.12b | 1.16a | 0.30b | 3.56b | 3.87b | 0.92d | 7.43b | 71.83b |

[z] Total content= ($Rb_1$+$Rb_2$+$Rb_3$+Rc+Rd+$Rg_3$+$Rh_2$+Re+Rf+$Rg_1$+$Rg_2$). [y] Different letters within a column indicate significant difference at $p < 0.05$ according to Duncan's multiple range test ($n = 3$).

### 3.5. MDA, CAT, and POD Activity of Mutant Roots

Antioxidase (POD and CAT) activities and MDA content were analyzed for clarify the distinctive characteristic in tetraploid and mutant roots. This allows a more detailed understanding of the molecular mechanism through which the mutagenesis effects on bioactive compounds stay obscure. MDA content was higher in mutant root lines than in the control line, and MDA contents were directly proportional to biomass which showed through the linear regression coefficients (r2) in the four mutant root lines (Figure 4). Mutant root lines 100–1-2 and 300–2-8 exhibited the lowest POD activities, whereas 100–1-18 and 300–1-16 activities were similar to those of the control. Mutant 100–1-2 also exhibited the lowest CAT value, whereas the other mutants resembled the control (Figure 5). No regression relationship among the POD and CAT actives and biomass was observed (data not shown). The colchicine-induced genetic alterations in the mutant lines may explain the differences in concentration and activities of some enzymes [40,41]. The difference in antioxidant enzyme activities and MDA content may be due to the upregulated expression of the corresponding genes.

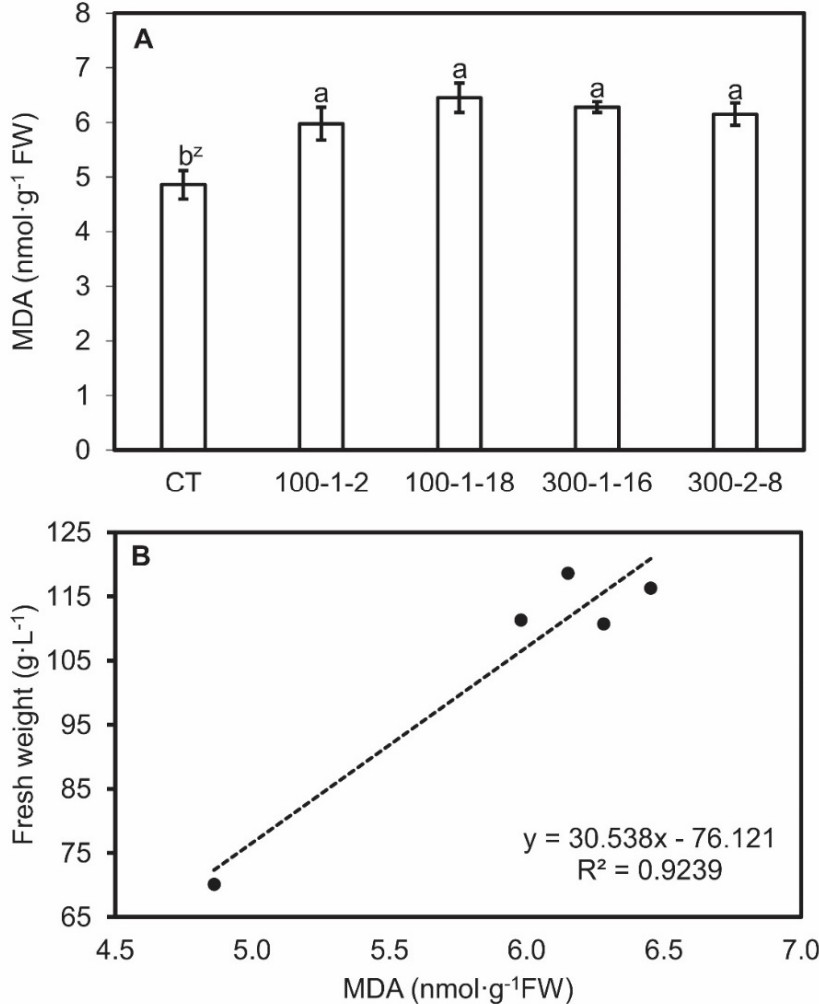

**Figure 4.** Malondialdehyde (MDA) content (**A**) and correlation between MDA content and fresh weight (**B**) in four mutant lines of *Panax ginseng* after six weeks of culture. CT, control. Data present the mean ± SE of three replications. $^z$ Different letters indicate significant difference at $p < 0.05$ according to Duncan's multiple range test ($n = 3$).

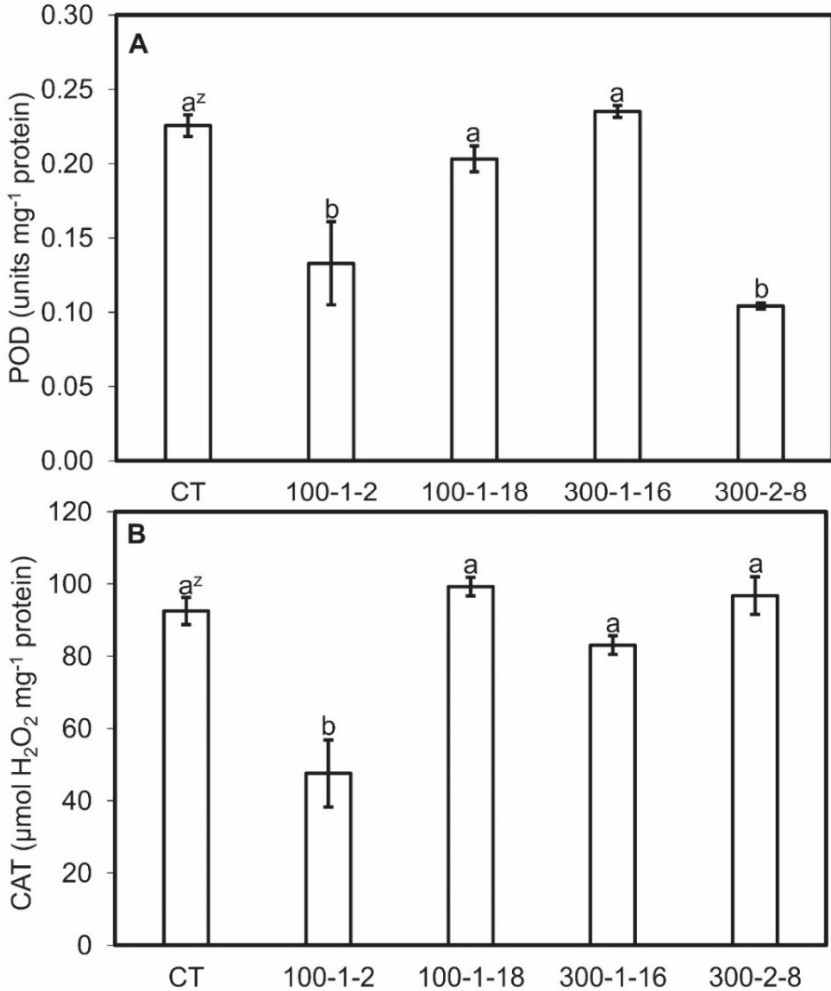

**Figure 5.** Peroxidase (POD) (**A**) and catalase (CAT) (**B**) activities in four mutant lines of *Panax ginseng* after six weeks of culture. CT, control. Data present the mean ± SE of three replications. [z] Different letters indicate significant difference at $p < 0.05$ according to Duncan's multiple range test ($n = 3$).

### 3.6. Bioreactor Culture of Mutant Lines

The selected mutant roots were scaled-up in a bioreactor system. All four mutant lines grew successfully in the bioreactor system during the six-week culture duration (Table 3; Figure 6). Mutant 300–2-8, which showed the highest fresh biomass, exhibited particularly rapid adaptation capacity in the bioreactor system. After six weeks of culture, lateral root number and length were highest in mutant 100–1-18 (Table 3). Overall, all four mutant lines were suitable for scaled-up cultivation in the bioreactor system, suggesting their amenability to commercial production at industrial levels. Mutagenesis led to increased vegetative and reproductive organ production in *P. ginseng*. Similarly, Nura et al. [21] showed that leaf and seed yield were improved in sesame mutants. Colchicine induction of mutagenesis in sesame caused genomic alterations that enhanced cell division and meristematic expansion [21].

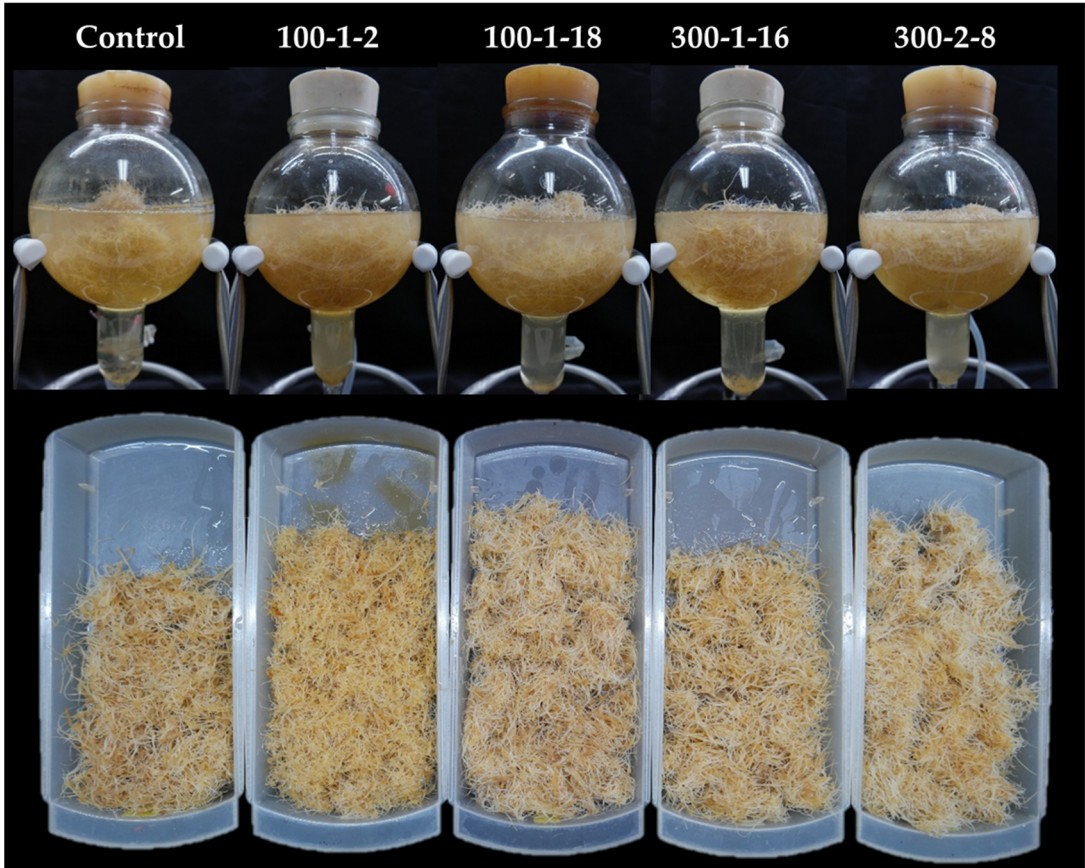

**Figure 6.** Scaled-up cultivation of four mutant lines in the bioreactor system.

## 4. Conclusions

Mutagenesis is a powerful strategy for inducing desirable, valuable features such as secondary metabolite production in plants and is particularly applicable to ginseng. Here, in vitro mutant induction by colchicine in long-term adventitious root cultures of *P. ginseng* is demonstrated. Mutant induction of wild ginseng adventitious roots is essential for improving the biomass and ginsenoside content in the ginseng in vitro culture systems. Mutant roots characterized by rapid growth and high production of secondary metabolites may improve the efficiency of industrial production systems for foods, pharmaceutical products, and cosmetic compounds, among others.

**Supplementary Materials:** The following are available online at http://www.mdpi.com/2073-4395/10/6/785/s1. Table S1: List of primers related to ginsenoside synthetic genes of *Panax ginseng*. Table S2. The retention time values of the ginsenoside standard of *Panax ginseng* by HPLC analysis. Figure S1: Flow cytometric histogram of nuclei of *Panax ginseng*.

**Author Contributions:** K.-C.L., T.-T.H. and J.-D.L. contributed to data acquisition and writing of the manuscript. K.-Y.P. participated in interpreting the data. S.-Y.P. made substantial contributions to the conception and design of this study, and of revising important intellectual content. All authors have read and agreed to the published version of the manuscript.

**Funding:** This work was supported by Korea Institute of Planning and Evaluation for Technology in Food, Agriculture, Forestry and Fisheries (IPET) through Advanced Production Technology Development Program, funded by Ministry of Agriculture, Food and Rural Affairs (Grant number 315013-4).

**Acknowledgments:** Authors were supported by Brain Korea (BK) 21 Plus Program through the National Research Foundation (NRF) of Korea.

**Conflicts of Interest:** The authors declare no conflict of interest.

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
