# Peer review of "Colchicine Mutagenesis from Long-term Cultured Adventitious Roots Increases Biomass and Ginsenoside Production in Wild Ginseng (Panax ginseng Mayer)"

_agronomy, doi:10.3390/agronomy10060785_

Round 1

Reviewer 1 Report

In my opinion, the work is well written and provides interesting new results in the field of medicinal plant biotechnology. Noteworthy is the significant increase in ginsenoside levels in mutated roots, especially in clone 100-1-18. I have a question. Namely, have you tried to shorten the gradient program in HPLC analysis? 150 minutes is a long time, even for HPLC. The HPLC chromatogram contained in the manuscript, in my opinion, is required, also with the Rt values of the determined secondary metabolites (ginsenosides) and their standards.

Author Response

With reference to the above, authors are thankful to the reviewers for their valuable comments on manuscript.  We have revised the manuscript in the light of reviewer’s comments and incorporated all the corrections in the revised manuscript. The corrections incorporated in revised text are highlighted in blue color. The revised manuscript has been reviewed again by native English speaker. Following are specific corrections incorporated in the revised manuscript.

Reviewer 1

In my opinion, the work is well written and provides interesting new results in the field of medicinal plant biotechnology. Noteworthy is the significant increase in ginsenoside levels in mutated roots, especially in clone 100-1-18. I have a question. Namely, have you tried to shorten the gradient program in HPLC analysis? 150 minutes is a long time, even for HPLC. The HPLC chromatogram contained in the manuscript, in my opinion, is required, also with the Rt values of the determined secondary metabolites (ginsenosides) and their standards.

Answer:

Thank you for your comment. We tried the shorter gradient program in HPLC analysis. However, the peaks were not separate clearly. We also found a few unknown peaks after 110th min. Therefore, we run the analysis for 150 min for further study to separate clear peak. We added the HPLC chromatograms and table for Rt of HPLC analysis as a supplementary data, and cited them in the text.

Reviewer 2 Report

The manuscript meets all the requirements for this type of work. The Abstract corresponds to the contents of the paper. Information in the manuscript is a valuable addition to current knowledge and is overall written well, clear, and concise. The manuscript has international relevance and scientific merit. The objectives and methods described are clear and concise. Data tables and figures are appropriate. Results and Discussion are presented well.

I have no comments or observations on the content of this manuscript.

Author Response

With reference to the above, authors are thankful to the reviewers for their valuable comments on manuscript.  We have revised the manuscript in the light of reviewer’s comments and incorporated all the corrections in the revised manuscript. The corrections incorporated in revised text are highlighted in blue color. The revised manuscript has been reviewed again by native English speaker. Following are specific corrections incorporated in the revised manuscript.

Reviewer 2

The manuscript meets all the requirements for this type of work. The Abstract corresponds to the contents of the paper. Information in the manuscript is a valuable addition to current knowledge and is overall written well, clear, and concise. The manuscript has international relevance and scientific merit. The objectives and methods described are clear and concise. Data tables and figures are appropriate. Results and Discussion are presented well.

 I have no comments or observations on the content of this manuscript.

Answer:

Thank you for your review and evaluate our manuscript.
